# Surface-Related Kinetic Models for Anaerobic Digestion of Microcrystalline Cellulose: The Role of Particle Size

**DOI:** 10.3390/ma14030487

**Published:** 2021-01-20

**Authors:** Michał Piątek, Aleksander Lisowski, Magdalena Dąbrowska

**Affiliations:** Department of Biosystems Engineering, Institute of Mechanical Engineering, Warsaw University of Life Sciences, Nowoursynowska 166, 02-787 Warsaw, Poland; aleksander_lisowski@sggw.edu.pl (A.L.); magdalena_dabrowska@sggw.edu.pl (M.D.)

**Keywords:** surface-related kinetics, anaerobic digestion, microcrystalline cellulose, modelling

## Abstract

In this work, for modelling the anaerobic digestion of microcrystalline cellulose, two surface-related models based on cylindrical and spherical particles were developed and compared with the first-order kinetics model. A unique dataset consisting of particles with different sizes, the same crystallinity and polymerisation degree was used to validate the models. Both newly developed models outperformed the first-order kinetics model. Analysis of the kinetic constant data revealed that particle size is a key factor determining the anaerobic digestion kinetics of crystalline cellulose. Hence, crystalline cellulose particle size should be considered in the development and optimization of lignocellulose pre-treatment methods. Further research is necessary for the assessment of impact of the crystalline cellulose particle size and surface properties on the microbial cellulose hydrolysis rate.

## 1. Introduction

Lignocellulose materials are abundant substrates for biogas and renewable energy production [1,2,3]. A better understanding of the key factors in the degradation process is required to increase the use of these materials for biogas production. Lignocellulose is a unique, recalcitrant structure of material plant cell walls. Its complex structure hinders microbial degradation. Further, cellulose is water insoluble and forms highly stable hydrogen-bonded crystalline fibers. These characteristics prevents its efficient utilisation in the anaerobic digestion process [4,5,6,7,8,9].

It is necessary to better understand the relationships between lignocellulose composition, degree of degradation, and degradation kinetics [10,11]. Moreover, there is still little information regarding the relationship between the kinetics of lignocellulose degradation and the structure of cellulose itself [12], particularly regarding the impact of crystallinity and enzyme adsorption sites [13]. The crystalline structure of lignocellulose hinders its degradation, and thus, the hydrolysis of these insoluble compounds is slow. However, the mechanisms behind the reduced hydrolysis rate are not fully understood; the low hydrolysis rate may be attributed to enzyme-associated factors such as enzyme inactivation, jamming due to enzyme overfilling on the cellulose surface, and inhibition of the enzymes. In addition, substrate-related factors, such as multiphase cellulose composition, changes in the degree of polymerization, crystallinity, reduction in reactivity, and substrate availability, as well as other physical properties may also contribute to the reduced hydrolysis rate [14].

In addition, the hydrolysis rate depends on many strongly related factors, such as the type and structure of the substrate (e.g., surface area and crystallinity), cellulose activity, and reaction conditions [15]. Some studies have shown that a decrease in crystallinity is invariably accompanied by a decrease in other substrate characteristics such as particle size and an increase in the available surface area that results from the pre-treatment of the material (mainly achieved by grinding) [16,17].

Hydrolysis of lignocellulose is considered to be the rate-limiting step in the anaerobic digestion process [11,18,19]. It has also been shown that the hydrolysis rate is controlled by enzyme kinetics if the amount of the enzyme exceeds the amount of the adsorption sites. In this case, hydrolysis can be described by first-order kinetics that have been widely used to describe the hydrolysis kinetics of cellulose and other solid particulate materials [20,21,22,23,24]. It is clear from the previous studies [21,22] that all accessible surface sites were used by enzymes; however, the first-order kinetics model does not describe a decrease in the amount of adsorption sites related to the shape and size of particulate matter. Furthermore, during anaerobic digestion, cellulose is hydrolyzed by cellulosomes that are multienzyme complexes produced by cellulose-degrading anaerobic microbes and mediate cell attachment to the of particulate [25,26,27]. After attachment to cellulose particle, cellulosomes grow rapidly and maintain contact with bacteria [26]; thus, enzyme–particle collisions can be necklaced as process rate limiting factor. Therefore, the available particulate surface is a crucially important factor. Some surface-related models of particulate hydrolysis have been developed [28,29,30], but this problem requires further investigation, particularly regarding cellulosome activity. To the best of our knowledge, surface-related models for anaerobic digestion have not been developed or improved upon in recent works. In this paper, we present a newly developed surface-related approach for the modelling of particulate matter degradation during anaerobic digestion using microcrystalline cellulose (MCC) as the model substrate. Pure cellulose was used in experiment to avoid interferences with lignin such as competitive and non-competitive inhibition [31]. The goal of this study was to isolate the role of cellulose particle size, so in our data set polymerization degree and crystallinity are also constant across samples, despite the fact that it is well known that mentioned above factors have significant impact on cellulose degradation kinetics [3]. It was assumed that the cellulosome activity on the entire accessible MCC surface was constant; however, particle surface, related to particle radius, decreased; thus, change in kinetics is strictly surface related. Based on this assumption, we developed two surface-related models for the hydrolysis of cylindrical and spherical particles, respectively, and compared these models to the first-order kinetics model. A unique dataset comprising of data for the particles with the same crystallinity and degree of polymerisation but with different sizes was used to validate the model. Such a dataset has not been used in previous studies. This constitutes an additional novelty in this paper. Due to a lack of reliable lag-phase-duration data pertaining to cellulosome formation as well as for the simplest possible modelling of the hydrolysis process activation, all of the tested models incorporated lag-phase modelling with the assumption of sudden hydrolysis process activation after the lag time.

## 2. Materials and Methods

### 2.1. Experimental Data

#### 2.1.1. Material Preparation

MCC Flocel 102 was purchased from a local market (ITC, Piastów, Poland) and was separated according to the ANSI/ASAE S319.4 standard using a LAB-11-200/UP sieve separator (Eko-Lab, Brzesko, Poland) with oscillatory motion in the vertical plane [32,33]. The set of sieves had dimensions of the opening screens in the following sequence from the bottom to the top: 20, 32, 45, 56, 100, 150, 212, and 300 μm. A pan was located at the bottom. The sieve set was selected based on the initial test such that approximately half of the material passed through the middle mesh size. The mass of a single sample of cellulose subjected to separation was 50 g. The duration of sieving was 300 s and was controlled using a stopwatch. The end-point was determined based on preliminary experiments. For time durations between 300 and 600 s, the mass on the pan did not change by 0.1%; therefore, for economic reasons, it was decided to set the duration as 300 s. Seven samples of particle fractions between 20 and 300 μm were obtained. Mean geometric size of particles were calculated as geometric mean of top and bottom sieve size of given pair of sieves.

#### 2.1.2. Anaerobic Digestion

The samples were processed by anaerobic digestion. The inoculum was obtained from a mesophilic, agricultural biogas plant post-digester. Inoculum contained 3.1% ± 0.05% total solids and 63.2% ± 0.05% volatile solids. Each fraction (10 g) was weighed on a WPS 600/C electronic scale (Radwag, Radom, Poland) with an accuracy of 0.01 g and placed into 2 dm^3^ bottles. Following this, inoculum (1800 mL) was added. Bottles with inoculum only (1800 mL) were also prepared as blank. The inoculum-to-substrate ratio in the reactor was 3.5:1 based on organic dry matter in order to eliminate a possible negative effect of a high substrate dose on the process [34,35]. The bottles were flushed with nitrogen and placed in a water bath at 37 °C, and the reactor contents were manually mixed once a day. The amount of gas was measured by the brine displacement method [36]. Displaced brine was weighed on electronic scales with an accuracy of 0.01 g, as frequently as is necessary to fully recognize the course of gas formation (once a day or every second day at the end of process) [37]. Batch trials were simultaneously conducted in triplicate, and the obtained biogas volume was converted to the volume at standard conditions (1013 hPa, 273 K). The anaerobic digestion process was conducted for 25 days. Anaerobic digestion of the particle fraction in the 45–56 μm range was repeated because of trial failure on the first approach. Inoculum from the same biogas plant was used, but it was collected at a different time. Test results for particle fraction in the 45–56 μm range are given herein; however, as they were not obtained under the same conditions, they are not included in the overall analysis.

#### 2.1.3. SEM Imaging

Images of randomly chosen microcrystalline cellulose samples were obtained by scanning electron microscopy (SEM, FEI Quanta 200 with microanalyser model EDS and digital image record; FEI Company, Hillsboro, OR, USA; merged with Philips Electron Optics) at the Analytical Centre of WULS. To obtain SEM images, a few cellulose particles were taken from each fraction and placed into a vacuum chamber.

#### 2.1.4. True Density Measurement

Particle density was measured via gas stereopycnometry (Quantachrome Instruments, Boynton Beach, FL, USA). A mass sample *m* of MCC was placed in the measuring cell with a capacity of *V_C_* = 100 cm^3^. For this state, the pressure *p*_1_ of the helium gas in the measuring cell was recorded. Then, the valve through which the gas was directed to the reference cell with the volume *V_A_* was opened, and its pressure *p*_2_ was measured [38]. The true density of the MCC was determined as
(1)$$\rho =\frac{m}{V_{C}-V_{A}/\left(\frac{p_{1}}{p_{2}}-1\right)},$$
where *ρ* is the true density (g·cm^−3^), *m* is the mass of the MCC sample (g), *V_C_* is the volume of the measuring cell (cm^3^), *V_A_* is the reference cell volume (cm^3^), *p*_1_ is the pressure in the measuring cell (MPa), and *p*_2_ is the reference cell pressure (MPa).

#### 2.1.5. Measurement of the Polymerization Degree

The polymerization degree was measured according to the PN-90/E-04421 and PN-92/P-50101/01 standards. Solution viscosity was assessed using an Ubbelohde viscometer (Schott Instruments GmbH, Mainz, Germany) with a capillary diameter of 0.84 mm. The cellulose solution outflow time was recorded every 10 min for 2 h for each sample weighing approximately 0.4 g. The capillary temperature was 25 ± 0.1 °C, and the error of the limiting viscosity number did not exceed 2%. The polymerization degree was calculated using the Immergut equation, *DP*^α^ = *K*[η], where *DP* is the average degree of viscosity polymerization (–), [η] is the weight average intrinsic viscosity (–), and *K* and α are the empirical constants equal to 1.65 g·cm^−3^ and 0.9, respectively [39]. To minimize the deleterious effect of the copper (II) ethylenediamine (CUEN) solution, the polymerization degree for zero retention time was calculated by extrapolation.

#### 2.1.6. Crystallinity Index Measurement

The crystallinity of the MCC samples (approximately 0.5 g) was evaluated by X-ray diffractometry (Tur M62 with a horizontal goniometer HZG-4, Carl Zeiss AG, Jena, Germany) with a Cu Kα radiation source (wavelength *λ* = 1.5418 Å) at a voltage of 30 kV and current of 25 mA. The scanning scope was 2θ = 5–30° with a step size of 0.04°, and the impulse counting time was 3 s. The crystalline structure of MCC gives rise to distinct peaks at 15°, 16.4°, and 22.5°. The cellulose crystallinity was calculated by the XRD deconvolution method [40,41]. From the peak deconvolution method, the amorphous peak (2θ) was predicted to be located at approximately 21°.

#### 2.1.7. Specific Surface Measurement

Specific surface area (SSA) was measured using the Brunauer–Emmett–Teller (BET) method [42]. Samples were degassed in vacuum for 24 h in 105 °C prior to measurement in order to remove water and other contaminations. Measurements were performed using a QUADRASORB SI surface area analyzer (Quantachrome Instruments, Boynton Beach, FL, USA). Nitrogen gas was used as an adsorbate. All the necessary calculations were made using the 3 Flex analyzer software (v. 3.01, Micromeritics Instr., Norcross, GA, USA).

#### 2.1.8. Water Retention Value Measurement

Water retention value (*WRV*) was measured according to the previously used protocol [43,44]. A precisely weighted sample (MA 50/1R electronic scale, Radwag, Radom, Poland with an accuracy of 0.1 mg) of cellulose (1 g) was placed in a centrifuge tube containing deionised water (10 mL), mixed, and left for 24 h. Then, the samples were centrifuged (3000× *g* for 20 min), and the supernatant was decanted. The remaining wet MCC samples were weighted and dried in an oven for 10 h in 105 °C. Then, the water retention value (*WRV*) was calculated as
(2)$$WRV=\frac{W_{wet}-W_{dried}}{W_{dried}},$$
where *WRV* is the water retention value (g H_2_O∙g^−1^), *W_wet_* is the weight of the wet centrifuged sample (g), and *W_dried_* is the sample weight after drying (g).

### 2.2. Kinetics Models

#### 2.2.1. The First-Order Kinetics Model

In this study, three types of kinetic models were compared. The first was the first-order kinetics model with a lag phase [45] given by
(3)B(t)=BP⋅(1−exp(−k⋅(t−λ))),
where *B*(*t*) is the biogas production at time *t* (mL·g^−1^), *BP* is the potential ultimate biogas production (mL·g^−1^), *t* is the time (h), *k* is the first-order kinetics constant, and *λ* is the duration of the lag phase (h).

Sudden process activation after the lag phase was assumed. Thus, the lag during the lag time kinetics constant *k* is 0; the model is first deactivated, and then activated when *t*-*λ* is higher than 0. For the first-order kinetics model, the half hydrolysis time is expressed as
(4)$$t_{0.5}=ln(2)/k,$$
where *ln*() denotes natural logarithm.

As mentioned above, this model focuses on the kinetics of substrate degradation with a kinetic constant *k*. It was found that the kinetic constant has different values for different substrate sizes even though this process occurs under the same pH and temperature conditions [21].

The SEM micrographs are presented in Figure 1. The SEM analysis of the shape of the MCC particles indicates that particle shape depends on the size of the mixture fraction separated by the sieves. Omitting surface irregularities and crystal roughness, it was assumed that the finer particles in the size range of 20–45 μm have a cylindrical shape and can be approximated as cylinders. In contrast, the particles with the sizes in the range of 56–212 μm form agglomerates in which the particle geometry can be approximated as spherical. These observations inspired the development of two mathematical models in which the access of enzymes to particles depends on the shape of the latter. Mathematical descriptions of the kinetics of the surface-related hydrolysis for the cylindrical and spherical particle shapes are presented below.

#### 2.2.2. Derivation of Surface-Related Models

A basic surface-related model equation was previously presented in the literature [46] as
(5)$$\frac{dM}{dt}=-k_{s}\cdot S$$
where *M* is the mass of the substrate (g), *t* is the time (h), *S* is the surface susceptible to hydrolysis (cm^2^), and *k_s_* is the surface-related hydrolysis constant (g·cm^−2^·h^−1^); however, detail derivation of model assumptions was not presented there. Step by step derivation is as follows. 

It is possible to rewrite the above equation in terms of the released volume of the substrate, obtaining
(6)$$\frac{dV}{dt}=\frac{-k_{s}\cdot S}{\rho }$$
where *V* is the substrate volume (cm^3^), and *ρ* is the substrate density (g·cm^−3^).

Considering the degradation of spherical particles, the change in the volume can be expressed as
(7)$$\frac{dV}{dt}=\frac{-k_{s}\cdot 4\cdot \pi \cdot R^{2}}{\rho },$$
where *R* is the particle radius (cm). 

We note that
(8)$$\frac{dV}{dt}=\frac{dV}{dR}\cdot \frac{dR}{dt}=4\cdot \pi \cdot R^{2}\cdot \frac{dR}{dt},$$

Thus,
(9)$$4\cdot \pi \cdot R^{2}\cdot \frac{dR}{dt}=\frac{-k_{s}\cdot 4\cdot \pi \cdot R^{2}}{\rho },$$

This simplifies to
(10)$$\frac{dR}{dt}=\frac{-k_{s}}{\rho }.$$

Thus, for spherical particles, it is possible to construct a model assuming an isotropic substrate and a constant decrease in the radius. It can be concluded that the rate of reaction is proportional to the particle surface area; thus, the decrease in the particle radius is constant. It was assumed that for cellulose, only the external layer of material is hydrolyzed at a given time. This approach is in line with the approach used for modelling the surface-related degradation of starch [46]. The present assumption for spherical particles is also valid for cylindrical particles with the assumption of radial particle degradation. Models sharing presented assumptions, adapted for biogas production curves, were not presented in literature before. In next sections full derivation of models adapted for biogas production curves are presented.

#### 2.2.3. Model for Cylindrical Particles

Based on the presented assumptions for degradation kinetics, the model for cylindrical particles is expressed as
(11)$$B\left(t\right)=BP\cdot \rho \cdot n\cdot \pi \cdot L\cdot (R^{2}-\left(R-\frac{k_{s}}{\rho }\cdot t{)}^{2}\right),$$
where *B*(*t*) is the cumulative biogas yield at time *t* (mL·g^−1^), *BP* is the ultimate biogas production (mL·g^−1^), *n* is the number of particles, *L* is the average particle length (cm), and *R* is the initial average particle radius (cm). In this model, measured particle radius and length can be introduced; however, knowledge about particles number is then necessary. 

The introduced lag phase model takes the following form
(12)$$B\left(t\right)=BP\cdot \rho \cdot n\cdot \pi \cdot L\cdot (R^{2}-\left(R-\frac{k_{s}}{\rho }\cdot \left(t-\lambda \right){)}^{2}\right).$$

The model shares lag-phase assumptions with the first-order kinetics model. If biogas or biomethane production curves are available without any additional information about the substrate specific surface or density, the only viable approach for modelling using surface-related models is relative modelling conducted with the assumption that the volume of the particle integrates to one. Additionally, assuming particles homogeneity, density is equal across particles. This allows the omission of the substrate density and the number of particles in the models. This makes models more clear in interpretation. Therefore, the model for the relative modelling of degradation for cylindrical particles is expressed as
(13)$$B\left(t\right)=BP\cdot \pi \cdot L\cdot (R^{2}-\left(R-r_{s}\cdot t{)}^{2}\right)$$
and
(14)$$B\left(t\right)=BP\cdot \pi \cdot L\cdot R^{2}\cdot (1-\left(1-\frac{r_{s}}{R}\cdot t{)}^{2}\right),$$
where *r_s_* is the kinetic constant for relative modelling that considers the decrease in the particle radius. The unit of *r_s_* is cm·h^−1^. It can be assumed that the total particle volume is equal to 1, ensuring that the most common structure of empirical equations is used for biogas or methane cumulative production curve modelling. Here, the constant related to the gas production potential is multiplied by an expression related to process dynamics. The dynamics-related expression integrates to 1 in its domain in nearly all cases [47]. It is assumed that the volume of a cylinder is equal to 1
(15)$$\pi \cdot L\cdot R^{2}=1.$$

Hence, Equation (14) can be transformed and simplified to
(16)$$B\left(t\right)=BP\cdot (1-\left(1-k\cdot t{)}^{2}\right),$$
where *k* is the first-order kinetic constant (h^−1^). Next, *λ* for lag-phase modelling is introduced, so that the final equation form is
(17)$$B\left(t\right)=BP\cdot (1-\left(1-k\cdot \left(t-\lambda \right){)}^{2}\right).$$

For Equation (17), the reciprocal of the constant *k* is the time required to complete the conversion after the lag phase. After this time, the constant *k* is assumed to be 0, as during the lag phase. The half decay time (*t*_0.5_) is given by
(18)$$t_{0.5}=\left(1-\sqrt{0.5}\right)/k.$$

#### 2.2.4. Model for Spherical Particles

The model for spherical particles is based on the same assumptions as the previous model, with the exception that the particles are spherical; therefore, the basic equation is
(19)$$B\left(t\right)=BP\cdot \rho \cdot n\cdot \frac{4}{3}\cdot \pi \cdot (R^{3}-\left(R-\frac{k_{s}}{\rho }\cdot t{)}^{3}\right).$$

Introducing a lag phase, the model takes the following form
(20)$$B\left(t\right)=BP\cdot \rho \cdot n\cdot \frac{4}{3}\cdot \pi \cdot (R^{3}-\left(R-\frac{k_{s}}{\rho }\cdot \left(t-\lambda \right){)}^{3}\right).$$

For relative modelling, the model takes the following form
(21)$$B\left(t\right)=BP\cdot \frac{4}{3}\cdot \pi \cdot (R^{3}-\left(R-r_{s}\cdot t{)}^{3}\right)$$
and
(22)$$B\left(t\right)=BP\cdot \frac{4}{3}\cdot \pi \cdot R^{3}\cdot (1-\left(1-\frac{r_{s}}{R}\cdot t{)}^{3}\right).$$

By analogizing to the model of cylindrical particles, it is possible to assume that volume of a sphere is equal to 1. Thus,
(23)$$\frac{4}{3}\cdot \pi \cdot R^{3}=1,$$
(24)$$B\left(t\right)=BP\cdot (1-\left(1-k\cdot t{)}^{3}\right).$$

Next, we include *λ* for lag-phase modelling; therefore, the final equation form is
(25)$$B\left(t\right)=BP\cdot (1-\left(1-k\cdot \left(t-\lambda \right){)}^{3}\right).$$

The reciprocal of the constant *k* is the time required to complete the conversion after the lag phase. The half decay time (*t*_0.5_) is given by
(26)$$t_{0.5}=\left(1-\sqrt[{3}]{0.5}\right)/k.$$

### 2.3. Model Fitting

All of the calculations were performed using the R package (version 3.5.1). The functions implemented in the “GenSA”, “minpac.lm” libraries were used to model biogas production. The models were fitted with the weighted least-squares method. The weights were calculated as a local slope of the biogas production curve [48]
(27)$$w_{i}=B_{i}-B_{i-1},$$
where *w_i_* is the weight factor (mL·g^−1^), *B_i_* (mL·g^−1^) is the biogas production curve value at a given measurement time *i*, and *B_i-_*_1_ (mL·g^−1^) is the previously measured value of the biogas production curve.

Two algorithms were used during the approximation process. The Levenberg-Marquardt (“nlsLM” function from “minpac.lm” library) algorithm was preceded by a generalised simulated annealing algorithm (“GenSA” function from “GenSA” library) to initiate parameter estimation [49]. The default algorithm parameters were used for both algorithms. The global relative error was used to evaluate the fit [50]
(28)$$\delta =100\sqrt{\overset{n}{\underset{i=1}{\sum }}{\left(z_{i}-p_{i}\right)}^{2}/\overset{n}{\underset{i=1}{\sum }}z_{i}{}^{2}},$$
where *n* is the number of measurements, and *z* and *p* are the actual values and values obtained from the model, respectively. Error was estimated for the whole curves and alternatively for the last day of the trial only, to emphasize error on biogas yield.

Because all the models used in this study had the same number of parameters, additional informational criteria (such as the Akaike criterion) were not determined.

### 2.4. Statistical Analysis

The Kruskal–Wallis test (“kruskal.test” form “stats” library) was used to compare the whole biogas production curves. The values of cumulative biogas production, cellulose moisture, and true density were compared using analysis of variance (“aov” function from “stats” library). Levene’s test for the homogeneity of variance (“leveneTest” function from “car” library) and the Shapiro–Wilk normality test (“shapiro.test” function from “stats” library) were used to check the validity of the analysis of variance assumptions. Differences were considered statistically significant at *p* ≤ 0.05 for the statistical test.

## 3. Results and Discussion

### 3.1. Material Characterisation

The SEM images of MCC are shown in Figure 1, and it can be concluded that this material is characterized by irregular shape and polydispersity [51]. However, there are significant differences between the shapes of the small particles and large particles, with the geometric mean particle sizes of 25–38 and 75–178 μm, respectively. These particle shapes resemble cylinders and spheres, respectively. The smallest particles of MCC were mostly independent, while large particles formed spontaneous agglomerates due to the aggregation of single, fine crystals. Individual MCC particles from kenaf bast and wood pulp [52] and from cellulose filter papers [53] were determined to be rod-shaped. The particle shape of Flocel MCC with a mean volume diameter of 153 μm was determined to be generally fibrous with a few plate-like structures [54]. The physicochemical properties of the material are presented in Table 1.

Significant differences were found for the true density of MCC particles (*p* = 0.0008), but the values varied in a very narrow range of 1.581–1.603 g·mL^−3^. There was no evidence regarding differences in moisture (*p* > 0.05). The MCC moisture content was in the narrow range of 4.89–5.20% w.b., and is comparable to the values of 3.96–5.06% w.b. obtained for other types of MCC (Avicel, Flocel, fine powder, Ranq, fibres from sisal) [54]. The values of sample crystallinity can be compared when the moisture contents of the samples do not differ significantly [55]. XRD showed comparable results for crystallinity in the range of 56–59% (without the 45–56 μm particle fraction). In addition, the results for the polymerisation degree were comparable in a range of 328–336 (Table 1). Four types of MCC, namely, Avicel, Flocel, fine powder, and fibres from sisal, were obtained and had similar values of 60% for the crystallinity index (calculated by the Segal formula) [54]. The lack of correlation (r = –0.028) between the crystallinity and particle size differs from other findings in the literature [56]. However, it should be emphasized that the split of the mixture into fractions is not caused by the grinding of the material during which particle crushing occurs [56]. In this study, the finer particles merged into spontaneous agglomerates, and during this process, the fine particles did not undergo substantial physical changes. This indicates that the polymerization degree and material crystallinity do not impact the sample kinetics comparison. This conclusion is in line with previous studies that indicated that crystallinity is only one of several parameters that should be taken into account when assessing the enzymatic rate of cellulose degradation [57]. Further, relatively small changes in the crystallinity index should not be correlated with the changes in cellulose digestibility [41]. The results of the specific surface and *WRV* measurements are presented in Table 2.

Statistically important differences in *WRV* were observed (*p* < 0.05). While the *WRV* values for the samples varied in a narrow range of 2.5–3.4 g H_2_O·g^−1^, the differences were distributed across the whole range of particle diameter. Similarly, specific surface results varied only in the 0.73–0.87 m^2^·g^−^^1^ range (Table 2) and were distributed across a range of diameters. In Section 3.3., the results presented in this section are analyzed in detail along with modelled data described in Section 3.2.

### 3.2. Results of Anaerobic Digestion and Model Comparison

Biogas production potential values are summarized in Table 3. Obtained biogas yield allowed to conclude that hydrolysis was completed, and theoretical biogas production potential was recovered [37]. Based on the Kruskal–Wallis test, significant differences (*p* = 0.0105) were found between the biogas production curves. Therefore, the curves were modelled separately. Biogas production curves are depicted in Figure 2, along with the model approximations for each MCC fractional sample.

However, analysis of variance did not provide statistically significant evidence for the differences in cumulative biogas production (*p* = 0.0622). A comparison of the results of these two tests indicated that the differences between the biogas production curves can be attributed to different process kinetics.

Comparison of lag-phase duration, half decay time, and complete decay time is presented in Table 4. All models approximated the biogas production curves well. The obtained values for the lag-phase duration, half hydrolysis time, and complete hydrolysis time are presented. Lag-phase duration can be considered as an effect of whole population or single cell growth [58]. As previously mentioned, during the batch tests, the inoculum-to-substrate ratio was 3.5:1 (based on organic dry matter); thus, the system was saturated with microorganisms. Therefore, the observed lag phase cannot be related to microbial population growth. The apparent lag should be considered as the time required for the synthesis of cellulosomes by individual cells and subsequent hydrolysis start-up. For the particles with the size in the 45–56 μm range, the lag-phase duration and half decay time were the longest, which was consistent for all of the tested models (Table 4). 

This indicates that the inoculum used for anaerobic digestion of this sample, which was collected at a different time and had a lower activity, cannot be directly compared to the rest of the data. For this reason, the results for the particles with sizes in the 45–56 μm range were excluded from the further analysis of the lag phase, decay time, and overall kinetics. The experimental and model values of lag-phase duration were consistent for the rest of the particles and on average were 2.93, 2.56, and 2.71 d for first-order, cylindrical, and spherical particle models, respectively. This confirms the assumption of sudden hydrolysis process activation after the lag phase. This approach enables the calculation of the complete process time by adding the lag-phase duration and the time for the completion of decay/hydrolysis. Only surface-related models enable the calculation of the time to complete hydrolysis.

Comparison of model performance has shown that process kinetics are surface related, which is in line with previous research on cellulose hydrolysis by cellulosomes [13]. The spherical particle model estimated a longer hydrolysis time than the cylindrical particle model, and the average times were 15.64 and 11.40 d (excluding particles with the sizes in the 45–56 μm range), respectively. The smallest and largest differences were found for the particle fractions in the 20–32 and 212–300 μm size ranges (excluding particles with the sizes in the 45–56 μm range), respectively, and were 3.61 and 4.72 d, respectively (Table 4). The half decay times were similar for all tested models, with the values in the narrow range of 2.43–2.98 d; thus, differences between the surface-related models were observed at the end of the process. 

The results shown in Figure 2 and the data presented in Table 3 show that the first-order kinetics model overestimates real biogas production, with the exception of the curve for the 20–32 μm particle fraction, and the first-order kinetics estimates do not fit into the area marked by the curve and its standard deviation (Figure 2), in contrast to both surface-related models. This demonstrates the better prediction of anaerobic fermentation by these two models; however, it is not possible to clearly distinguish if the cylindrical or spherical model is better. The first-order kinetics model also exhibited the largest global relative error (6.03%) and the end-of-trial relative error (8.94%) for the 212–300 μm particle fraction. In comparison, the largest end-of-trial errors were observed for the 20–32 μm particle fraction, and were 2.37% and 2.10% for the cylindrical and spherical particle models, respectively (Table 5). 

In most cases, the end-of-trial relative errors were slightly lower for the spherical particle model in than those obtained by the cylindrical particle model; the obtained values were 1.51% and 1.64% (excluding the fraction with the particle sizes in the 45–56 μm range), respectively, and can be attributed to the improved lag-phase modelling by the spherical particle model.

### 3.3. Analysis of Relations between Kinetic Constants, WRV and SSA

In this study, cellulose crystallinity and polymerization degree remained constant across samples; thus, *k* values were influenced by available adsorption sites related to substrate surface. For better analysis of *k* values, we proposed an empirical equation, which is the opposite Michaelis–Menten saturation equation
(29)$$k=1-\frac{k_{max}\cdot d}{K_{s}+d},$$
where *k* is measured kinetic constant value (h^−1^), *k_max_* is the maximum kinetic constant value (h^−1^), *K_s_* is the half saturation constant (μm), and *d* is the mean particle diameter (μm). 

The obtained global relative errors were 2.08% and 1.71% for cylinder and spherical shaped particles model, respectively. The results obtained using this approximation are presented in Figure 3.

The proposed model interpretation refers only to process initialization and is as follows: when particle diameter is ≈0, nearly all adsorption sites are exposed. Then, with the increase in the particle diameter, the initial exposure of the enzyme adsorption sites decreases, so that the process requires more time based on the model assumptions (lower *k* constant). In this particular case, it can be stated that the difference in the particle size was the sole reason for better exposure of adsorption sites. During pre-treatment, both cellulose crystallinity and particle size usually change [16,17]. This is also true for ball milling of MCC [59]. A previous study on cellulose sonication has indicated the possibility of substantial hydrolysis enhancement due to considerable particle size reduction (from 38 μm to < 0.40 μm) with only slight changes in crystallinity (12% decrease) and degree of polymerization (no change), thus proving that recrystallisation is not inevitable [60]. Another study showed a considerable particle size decrease (from 300–500 to 20 μm in length and from 10–20 to 2 μm in diameter) after 120 h of enzymatic hydrolysis, while cellulose crystallinity decreased by only 12% [12]. Additionally, 400–500 nm channels were observed on the particles’ surfaces after 48 h. Moreover, acid hydrolysis can lead to production of cellulose nanoparticles (size reduction from 45–53 μm to 53.5 ± 5.3 nm) with only 1.84% decrease in crystallinity, but a substantial decrease in the polymerization degree (from 466 to 215) [61]. Enzymatic hydrolysis can also lead to similar results, obtaining a size reduction from 45–53 μm to 36.5 ± 1.9 nm, polymerization degree decrease from 466 to 293, and 9.87% decrease in crystallinity [61]. Hence, it can be concluded that enzymatic hydrolysis causes only a 10–12% decrease in cellulose crystallinity, and a substantial decrease in the particle size. These observations are in line with our research, proving that change in the particle size only can enhance cellulose hydrolysis kinetics. This is confirmed by the recent studies on cellulosome activity that concluded that cellulosomes are particularly active for the smallest cellulose crystals (length ≤ ~70 nm) [62]. Such small particles are hardly attacked by free enzymes, possibly due to enzyme jamming [62,63]. However, for anaerobic digestion of lignocellulose, the influence of the particle size can be observed for much larger particles (0.15–1.70 mm) [64]. This can be attributed to the modification of the lignocellulosic matrix rather than the cellulose itself. 

The *WRV* methods can be applied in order to measure the adsorption surface area. Typically, larger *WRVs* are associated with better surface exposure [65], larger inner pore volume, and lower crystallinity [43,44]. In MCC hydrolysis, hydrophobicity can be considered to be the key factor [44], and hence lower *WRVs* can be interpreted to be more beneficial if the crystallinity remains constant, which is true for this study. Figure 4 shows a comparison of *WRV* and *k* constant data, and it is observed that *WRV* increases with increasing particle size. Differences in uncertainty measures used on Figure 4 and Figure 5 are the effect of different calculation software used. *WRV* changes can be attributed to the higher pore volume. However, *WRV* is correlated with the particle size (r = 0.822), and thus the *WRV* data cannot be interpreted without the knowledge of the particle size; however, it changes in a relatively narrow range (2.52–3.39 g H_2_O·g^−1^).

The BET specific surface area is frequently used to characterize lignocellulosic materials [43,66]. In this study, absolute SSA values do not differ substantially across samples, while kinetic constants show greater absolute differences, especially for the smallest particles. In this study, the surface area ranges from 0.73 to 0.87 m^2^·g^−1^ (Table 2). Surprisingly, the specific surface area of the 20–32 μm particles (0.80 m^2^·g^−1^) is lower than that of the 32–45 μm particles (0.87 m^2^·g^−1^). This can be attributed to the particle structure, because the 20–32 μm particles are rather individual particles, while the 32–45 μm particles are agglomerates of smaller particles (Figure 1). While some studies have indicated that SSA is the most important indicator of hydrolysis effectiveness [44], this conclusion was based on ball-milled cellulose samples, where SSA was altered simultaneously with other properties such as a substantial crystallinity change. Additionally, during hydrolysis by free enzymes, not only the surface area, but also the surface properties, such as roughness, are important and can cause enzyme jamming [67]. Comparison of SSA with *k* constants is presented in Figure 5. In this study, the SSA does not accurately describe the changes in process kinetics. In particular, the 20–32 μm particles show an opposite trend, because SSA decreases while *k* constant increases. Recent studies show that spatially organized enzymes in cellulosome can adapt their shape to cellulose nanocrystals. The individual cellulosome surface was calculated as approximately 1500 nm^2^ [62]. SSA measured using the BET method is due to the areas of mesopores that are defined as the pores with the widths in the 2–50 nm range [68]. These pores are likely to be smaller than the proposed cellulosome area; hence, according to presented data, it can be hypothesized that cellulosomes do not necessarily utilize the mesopore surface. Additionally, the formation of agglomerates leads to decreased adsorption sites accessibility [69].

## 4. Conclusions

All the tested models provide a good approximation of the hydrolysis process. However, the surface-related models outperform the first-order kinetics model that overestimates the true biogas production potential. The simple models presented in this work can be used as part of extended models in the evaluation of lignocellulose pretreatment methods. It is clear that the generation of amorphous cellulose during pretreatment is desirable because of its lower free energy demand for hydrolysis [44]. However, the unique data set presented in this paper and the newly developed surface-related modelling approach revealed that particle size is a key factor determining the kinetics of crystalline hydrolysis. Hence, crystalline cellulose particle size should be an important target property in the development and optimization of lignocellulose pre-treatment methods. Further research for the evaluation of the impact of the crystalline cellulose particle size and surface properties on the microbial cellulose hydrolysis rate is required.

## Figures and Tables

**Figure 1 materials-14-00487-f001:**
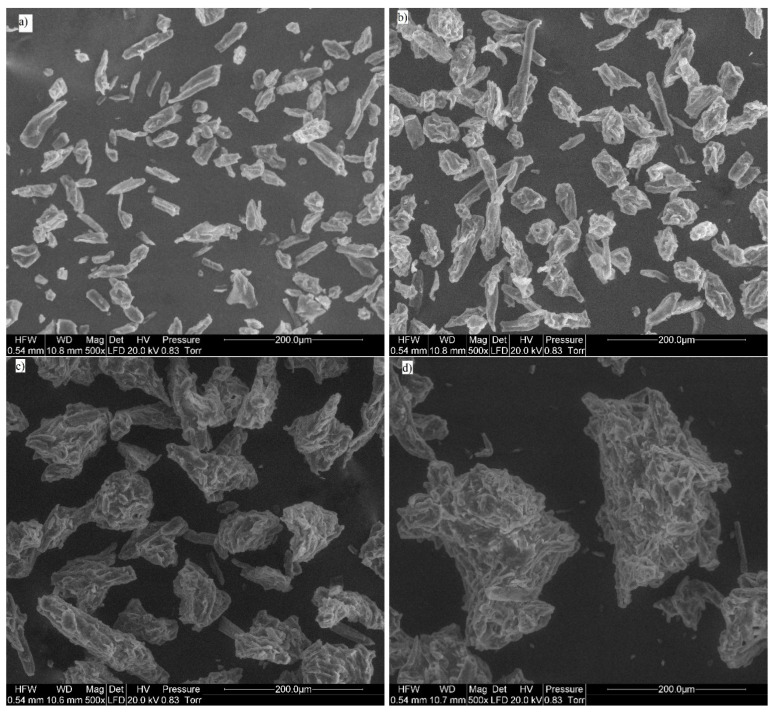
SEM micrographs of MCC particles with sizes of (**a**) 20–32 μm; (**b**) 32–45 μm; (**c**) 56–100 μm; (**d**) 150–212 μm.

**Figure 2 materials-14-00487-f002:**
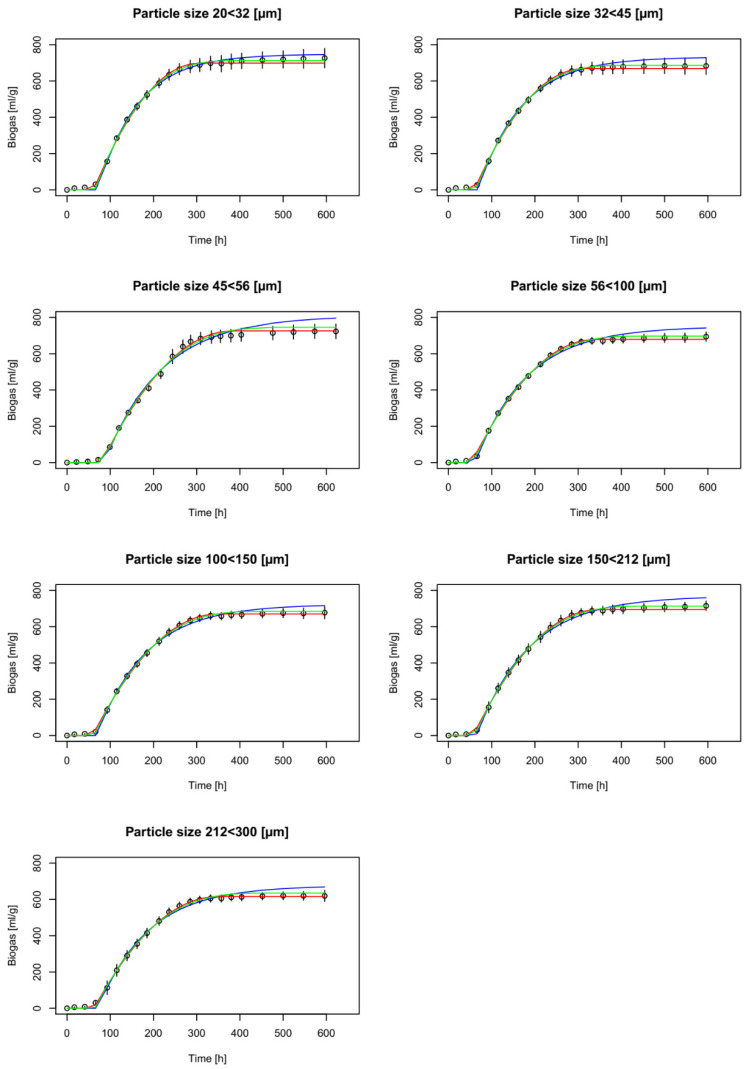
Biogas production experimental and model data for each sample. Circles denote the means of three samples, and vertical bars denote standard deviations. Blue, red, and green lines denote the first-order kinetic model and the approximations for the cylindrical and spherical particles, respectively.

**Figure 3 materials-14-00487-f003:**
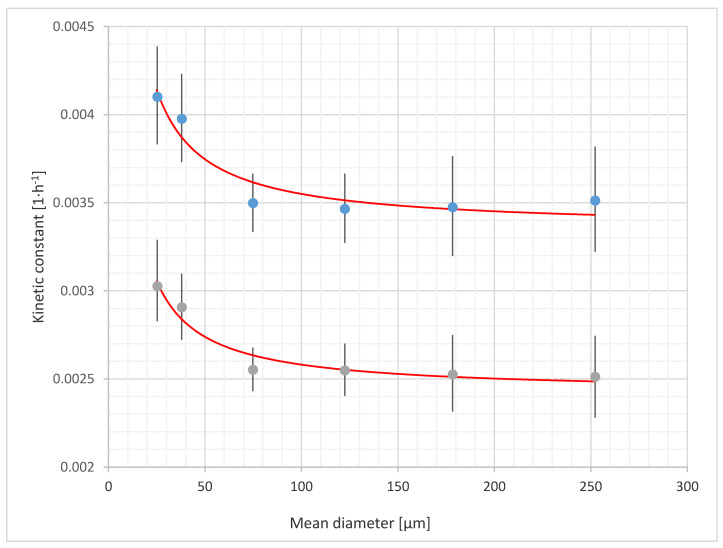
Measured *k* constants for the cylinder (top) and spherical (bottom) shaped particles models. Points represent measured values and lines represent model approximations.

**Figure 4 materials-14-00487-f004:**
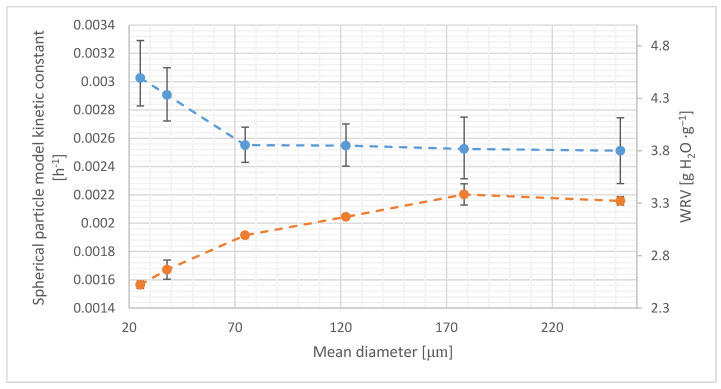
*WRV* compared to spherical shaped particle model kinetic constant values. For *k* constant (top) and *WRV* (bottom) points, the error bars represent the 95% confidence interval and standard error, respectively.

**Figure 5 materials-14-00487-f005:**
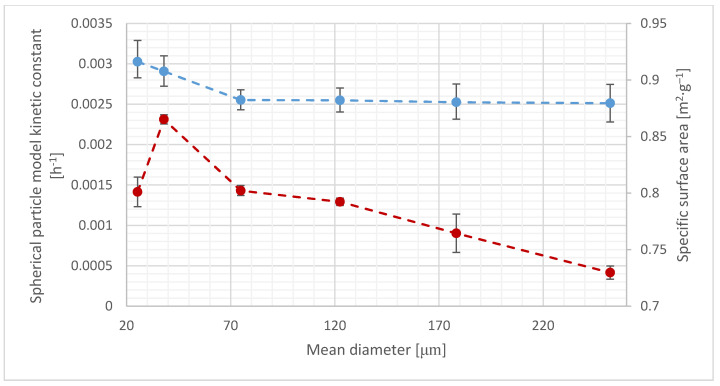
Specific surface area compared to the spherical shaped kinetic constant values. For the *k* constant (top) and SSA (bottom) data points, the error bars represent the 95% confidence interval and the standard error, respectively.

**Table 1 materials-14-00487-t001:** Basic material characteristics.

Fractions [μm]	Geometric Mean Particle Size [µm]	Moisture Content w.b.		True Density		Polymerisation Degree	Crystallinity
		Mean n = 3 [%]	SD [%]	Mean n = 3 [g·mL^−3^]	SD [g·mL^−3^]	Value n = 1 [–]	Value n = 1 [%]
20 < 32	25	5.20	0.28	1.597	0.001	328	56
32 < 45	38	4.89	0.16	1.600	0.001	336	58
*45 < 56*	*50*	*5.09*	*0.13*	*1.595*	*0.001*	*330*	*60*
56 < 100	75	4.97	0.32	1.589	0.005	333	58
100 < 150	122	4.90	0.35	1.581	0.009	335	59
150 < 212	178	5.05	0.08	1.603	0.001	333	57
212 < 300	252	4.95	0.29	1.593	0.006	333	57

SD—standard deviation.

**Table 2 materials-14-00487-t002:** Results of specific surface and *WRV* measurements.

Sample	Specific Surface Area		*WRV*	
	Mean n = 1 [m^2^·g^−1^]	Standard Deviation [m^2^·g^−1^]	Mean n = 5 [g _H2O_·g^−1^]	Standard Deviation [g _H2O_·g^−1^]
20 < 32	0.801	0.013	2.52	0.04
32 < 45	0.865	0.004	2.67	0.09
*45 < 56*	*0.834*	*0.002*	*2.72*	*0.05*
56 < 100	0.802	0.004	3.00	0.02
100 < 150	0.792	0.003	3.17	0.02
150 < 212	0.765	0.017	3.39	0.10
212 < 300	0.730	0.006	3.32	0.04

**Table 3 materials-14-00487-t003:** Comparison of measured and estimated biogas production potential.

Fractions [μm]	Measured Biogas Production Potential		First-Order Kinetic Model Estimates [mL·g^−1^]	Cylindrical Particle Model Estimates [mL·g^−1^]	Spherical Particle Model Estimates [mL·g^−1^]
	Mean [mL·g^−1^]	Standard Deviation [mL·g^−1^]			
20 < 32	726	54	748	698	712
32 < 45	683	47	732	668	686
*45 < 56*	*723*	*41*	*810*	*726*	*746*
56 < 100	694	27	750	679	697
100 < 150	677	34	723	670	684
150 < 212	715	26	769	694	713
212 < 300	619	31	676	615	635

**Table 4 materials-14-00487-t004:** Comparison of lag-phase duration, half decay time, and complete decay time. Complete decay time compared for surface related models only.

Fractions [μm]	First-Order Kinetic Model			Cylindrical Particle Model			Spherical Particle Model		
	Lag Phase [d]	Half Decay Time [d]	Complete Decay Time [d]	Lag Phase [d]	Half Decay Time [d]	Complete Decay Time [d]	Lag Phase [d]	Half Decay Time [d]	Complete Decay Time [d]
20 < 32	2.97	2.66	–	2.54	2.98	10.16	2.69	2.84	13.77
32 < 45	2.86	2.90	–	2.47	3.07	10.48	2.62	2.96	14.33
*45 < 56*	*3.59*	*3.79*	*–*	*3.32*	*3.68*	*12.57*	*3.45*	*3.60*	*17.44*
56 < 100	2.56	3.44	–	2.21	3.49	11.91	2.37	3.37	16.33
100 < 150	2.87	3.27	–	2.45	3.52	12.02	2.62	3.37	16.35
150 < 212	2.69	3.51	–	2.36	3.51	11.99	2.52	3.40	16.50
212 < 300	2.98	3.34	–	2.57	3.47	11.86	2.70	3.42	16.58

**Table 5 materials-14-00487-t005:** Comparison of model performance.

Fractions [μm]	First-order Kinetics Model		Cylindrical PParticle Model		Spherical Particle Model	
	Global Relative Error [%]	End-of-Trial Relative Error [%]	Global Relative Error [%]	End-of-Trial Relative Error [%]	Global Relative Error [%]	End-of-Trial Relative Error [%]
20 < 32	5.41	6.60	5.31	2.37	5.06	2.10
32 < 45	5.87	8.70	4.66	2.00	4.52	1.88
*45 < 56*	*7.49*	*11.09*	*5.05*	*1.53*	*5.59*	*1.84*
56 < 100	4.92	7.55	3.04	1.26	3.10	1.05
100 < 150	4.75	7.05	3.44	1.40	3.59	1.40
150 < 212	5.31	6.92	4.00	1.40	3.90	1.02
212 < 300	6.03	8.94	4.10	1.38	4.66	1.59

## Data Availability

The data presented in this study are available on request from the corresponding author.

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
