# Peer review of "Surface-Related Kinetic Models for Anaerobic Digestion of Microcrystalline Cellulose: The Role of Particle Size"

_materials, 2021, doi:10.3390/ma14030487_

Round 1

Reviewer 1 Report

The manuscript is well written, and the models come up with an interesting point about particle size as an important factor in the anaerobic digestion of microcrystalline cellulose. Some suggestions to improve the manuscript for readers follow.

2. Materials and Methods

2.1 Experimental data

2.1.1 Material preparations

In the second line of this section the authors refer to the ANSI/ASAE-S319.4. For readers who are not familiar as to what this is, can the authors provide a reference?

2.1.2 Anaerobic digestion

In the middle of the paragraph, the authors refer to the brine replacement method. For readers who may not know what this is, can the authors provide a reference?

The authors refer to measuring the amount of gas. How often this was done needs to be indicated.

2.1.4 True density measurement

The symbol for true density in formula (1) does not appear to match very well the symbol used in the line below the formula where true density is defined. Is this a typographical error?

2.1.5 Measurement of the polymerization degree

Can the authors provide a reference for the Brunaeur-Emmett-Teller method?

2.2.2 Derivatives of surface related models

The symbol for substrate density in formula (6) does not appear to match very well the symbol used in the line below the below the formula where substrate density is described. Once again, is this a typographical error?

3. Results and Discussion

3.1 Material characterization

Page 8, sentence starting 8 lines from the bottom: 'This conclusion is in line with previous studies that indicated that crystallinity is only one of several parameters that should be taken into account when assessing the enzymatic rate of cellulose degradation.' Can the authors indicate which previous studies they are referring to? 

Author Response

Dear Reviewer,

Thank you for your valuable comments. Now, the article is revised according to them and was improved. I added some references as requested, however I used Mendeley software, which does not synchronize with “track changes” option in MS Word, but pages and lines numbers are indicated in my responses. My responses to your comments are listed below.

[R:] 2. Materials and Methods

2.1 Experimental data

2.1.1 Material preparations

In the second line of this section the authors refer to the ANSI/ASAE-S319.4. For readers who are not familiar as to what this is, can the authors provide a reference?

[A:]I added reference (page 2 line 85)

[R:] 2.1.2 Anaerobic digestion

In the middle of the paragraph, the authors refer to the brine replacement method. For readers who may not know what this is, can the authors provide a reference?

[A:]I added reference (page 3 line 104)

[R:] The authors refer to measuring the amount of gas. How often this was done needs to be indicated.

[A:]I added notification in line with VDI 4630 standard: “Displaced brine was weighed on electronic scales with an accuracy of 0.01 g, as frequently as is necessary to fully recognize the course of gas formation (once a day or every second day at the end of process) [37]..  ” (page 3 line 107)

[R:] 2.1.4 True density measurement

The symbol for true density in formula (1) does not appear to match very well the symbol used in the line below the formula where true density is defined. Is this a typographical error?

[A:]Yes, it was. Corrected (page 3 line 126)

[R:] 2.1.5 Measurement of the polymerization degree

Can the authors provide a reference for the Brunaeur-Emmett-Teller method?

[A:]I added reference (page 4 line 150)

[R:] 2.2.2 Derivatives of surface related models

The symbol for substrate density in formula (6) does not appear to match very well the symbol used in the line below the below the formula where substrate density is described. Once again, is this a typographical error?

[A:]Yes it was. Corrected (page 6 line 195)

[R:] 3. Results and Discussion

3.1 Material characterization

Page 8, sentence starting 8 lines from the bottom: 'This conclusion is in line with previous studies that indicated that crystallinity is only one of several parameters that should be taken into account when assessing the enzymatic rate of cellulose degradation.' Can the authors indicate which previous studies they are referring to? 

[A:]I added reference to the review article (page 9 line 308)

Kind regards,

Michał Piątek

Reviewer 2 Report

This manuscript reports on the particle size dependence of anaerobic digestion of microcrystalline cellulose. The authors have conducted experiments by assigning detailed conditions to deepen the theoretical analysis. The presentation of the paper appears to be good. However, this paper contains several essential problems and I request the authors to corrected them before it can be considered for publication.

  1. Why did the authors use microcrystalline cellulose, which does not contain lignin, as a sample, even though they seemed to be interested in hydrolysis of lignocellulose (as stated in the introduction)? The introduction and the content of the experiment seem to be inconsistent.
  1. On page 8, the SEM image in Figure S1 should be included in the text to help the reader understand.
  2. In the specific surface area section of Table 2, there is almost no change in specific surface area even when the particle size is changed tenfold. Can you really say that the effect of specific surface area on enzymatic degradation has been detected?
  3. In Figure 3 and Figure 4, the authors try to contrast the left and right axes of the graphs, respectively. If that is the case, they should show the correlation between them directly in a graph with their axes on the x-y axis.
  4. After all, what approximations did the spherical and cylindrical approximations give to the MCC? The differences in the fitting of Fig. 1 do not seem to be that large. The trends in Figures 3 and 4 appear to show large differences because the vertical scale is stretched, but in terms of actual absolute values, there is not much difference, is there?

Author Response

Dear Reviewer,

Thank you for your opinion and valuable comments. Now, the article is revised according to them and was improved. My answers to your comments are listed below.

[R:] Why did the authors use microcrystalline cellulose, which does not contain lignin, as a sample, even though they seemed to be interested in hydrolysis of lignocellulose (as stated in the introduction)? The introduction and the content of the experiment seem to be inconsistent.

[A:]I added explanation: “Pure cellulose was used in experiment to avoid interferences with lignin such as competitive and non-competitive inhibition [31]. The goal of this study was to isolate the role of cellulose particle size, so in our data set polymerization degree and crystallinity is also constant across samples, despite the fact, that it is well known that mentioned above factors have significant impact on cellulose degradation kinetics [3].” (page: 2 lines 65-69:)

[R:] On page 8, the SEM image in Figure S1 should be included in the text to help the reader understand.

[A:]I moved SEM image from appendix to page 5 .

[R:] In the specific surface area section of Table 2, there is almost no change in specific surface area even when the particle size is changed tenfold. Can you really say that the effect of specific surface area on enzymatic degradation has been detected?

[A:]It is a good question and I believe it was already addressed. The answer is: not using BET. Issue of using BET specific surface results was carefully addressed in discussion (page 15 lines 443-452).: “In this study, the SSA does not accurately describe the changes in process kinetics. In particular, the 20–32 μm particles show an opposite trend because SSA decreases while k constant increases. Recent studies show that spatially organised enzymes in cellulosome can adapt their shape to cellulose nanocrystals. The individual cellulosome surface was calculated as approximately 1500 nm2 [61]. SSA measured using the BET method is due to the areas of mesopores that are defined as the pores with the widths in the 2–50 nm range [67]. These pores are likely to be smaller than the proposed cellulosome area, hence according to presented data, it can be hypothesised that cellulosomes do not necessarily utilise the mesopore surface. Additionally, the formation of agglomerates leads to decreased adsorption sites accessibility [68].”

[R:] In Figure 3 and Figure 4, the authors try to contrast the left and right axes of the graphs, respectively. If that is the case, they should show the correlation between them directly in a graph with their axes on the x-y axis.

[A:]The correlation between mentioned factors across all samples is highly influenced by leverage samples, which makes correlation graphs illegible. I believe that used data presentation approach is more suitable and provide better possibility in-depth analysis.

[R:] After all, what approximations did the spherical and cylindrical approximations give to the MCC? The differences in the fitting of Fig. 1 do not seem to be that large. The trends in Figures 3 and 4 appear to show large differences because the vertical scale is stretched, but in terms of actual absolute values, there is not much difference, is there?

[A:]I added: “This demonstrates the better prediction of anaerobic fermentation by these two models, however it is not possible to clearly distinguish if cylindrical or spherical model is better.” (page 13 lines 372-373)

I agree, there is not much absolute difference. That is why it is difficult to directly connect BET results with kinetic constant values, as it was addressed in discussion mentioned in previous answer. I added explanation: “In this study absolute SSA values do not differ substantially across samples, while kinetic constants show greater absolute differences, especially for the smallest particles”. (page 15 lines 432-434)

Kind regards,

Michał Piątek

Reviewer 3 Report

Is there a reference for section 2.2.3 and 2.2.4?  Is it 40?  It seems that this model should not new results.  If so, then that should be clearly noted.  If not, this section should be shorter.

What should be the minimum end-of-trail error for the application?  What error ranges for the modeling is required?

In captions 3 and 4, "the error bars represent the 95% confidence interval and standard error, respectively."  Does that mean different error method were used for each curve.  Why?  Should they both be the 95% confidence interval?

What is the source of the local increase for the second point in Figure 4?

Author Response

Dear Reviewer,

Thank you for your opinion and valuable comments. Now, the article is revised according to them and was improved. My answers to your comments are listed below.

[R:] Is there a reference for section 2.2.3 and 2.2.4?  Is it 40?  It seems that this model should not new results.  If so, then that should be clearly noted.  If not, this section should be shorter.

[A:]It is a reference to section 2.2.2 as mentioned there. Our model share part of presented in Sanders (2000), however derivation was not presented in detail in Sanders (2000), so we have decided to describe it extensively in our paper.

I added: “), however detail derivation of model assumptions was not presented there. Step by step derivation is as follows.” (page 6 lines 191-192)

There are no references to 2.2.3 and 2.2.4., so whole development process was precisely described, with as much details as possible.

I added: “Models sharing presented assumptions, adapted for biogas production curves were not presented in literature before. In next sections full derivation of models adapted for biogas production curves are presented” (page 6  lines 208-210)

[R:] What should be the minimum end-of-trail error for the application?  What error ranges for the modeling is required?

[A:]We did not assume borderline error value in advance. Our reference was measurements standard error, as mentioned with additional explanation (page 13  from line 368). We simply assumed that model with lower error is better.

I also added: “the first-order kinetics estimates do not fit into the area marked by the curve and its standard deviation (Fig. 1), in contrast to both surface-related models. This demonstrates the better prediction of anaerobic fermentation by these two models, however it is not possible to clearly distinguish if cylindrical or spherical model is better.” (page 13 lines 372-373)

[R:] In captions 3 and 4, "the error bars represent the 95% confidence interval and standard error, respectively."  Does that mean different error method were used for each curve.  Why?  Should they both be the 95% confidence interval?

[A:] Differences in used uncertainty measures in figures are the effect of different calculation software used. I agree, that using confidence interwall for all measurements is a better approach, however it cannot be changed. I believe that using standard error instead of confidence when absolute values do no differ substantially, do not have noticeable impact on presented conclusions.

I added additional explanation in text to avoid misunderstandings: “Differences in uncertainty measures used on figures Fig. 3 and 4 are the effect of different calculation software used”  (page: 15 lines 426-428)

And comment about low differences in absolute values: “In this study absolute SSA values do not differ substantially across samples, while kinetic constants show greater absolute differences, especially for the smallest particles”. (page 15 lines 432-433)”

[R:] What is the source of the local increase for the second point in Figure 4?

[A:] We have tried to address this question, as good as we could already: “Surprisingly, the specific surface  area of the 20–32 μm particles (0.80 m2·g–1) is lower than that of the 32–45 μm particles (0.87 m2·g–1). This can be attributed to the particle structure, because the 20–32 μm particles are rather individual particles, while the 32–45 μm particles are agglomerates of smaller particles (Fig. 1). “ (page 15 line 438)

Kind regards,

Michał Piątek

Round 2

Reviewer 2 Report

I think the authors have addressed all the issues.